# Parent-infant observation for prediction of later childhood psychopathology in community-based samples: A systematic review

**Elena McAndie**[1]\*, **Charlotte Alice Murray**[2], **Philip Wilson**[2], **Lucy Thompson**[2]

**1** CAMHS, NHS Grampian, Aberdeen, Scotland, **2** Centre for Rural Health, University of Aberdeen, Inverness, Scotland

\* elena.mcandie@nhs.scot

## Abstract

### Background

Difficulties in parent-child interaction are easily observed and are a potential target for early intervention. This review aimed to assess the utility of current observational methods used to assess parent-child interactions–within the first year of life–and their ability to screen and identify children from low-risk samples most at risk of developing childhood psychopathology.

### Methods

Six bibliographic databases were searched, and reference lists screened. All peer reviewed papers studying the association between an independent observation of parent-child inter-action and later childhood psychopathology in community-based samples were included. Included studies were those recruiting from population or community-based birth cohort data, which we define as 'low-risk'. Studies based on populations known to have a diagnosis of psychiatric illness or developmental disorder, or at high genetic or environmental risk of being diagnosed with such disorder, were excluded. Results were synthesised qualitatively due to high heterogeneity.

### Results

**20,051** papers were identified, nine were included in this study. Childhood psychopathology was associated with fewer positive parent-infant interactions, lower parent vocalisation fre-quency and lower levels of adult speech and activity. Maternal sensitivity was inversely related to separation anxiety and oppositional defiant/conduct disorders were associated with lower shared look rates. Disruptive behaviour disorders were associated with higher frequency of child vocalisation.

**Data Availability Statement:** All relevant data are within the paper and its Supporting Information files. All published papers discussed in the paper are in the public domain.

**Funding:** The author(s) received no specific funding for this work.

**Competing interests:** I have read the journal's policy and the authors of this manuscript have the following competing interests: PW is a co-author on five of the papers included in the analysis of this systematic review. This does not alter our adherence to PLOS ONE policies on sharing data and materials.

## Conclusion

Assessment of parent-child interactions, particularly the level of maternal activity, may be an early indicator of later childhood psychopathology in low-risk samples. Further longitudinal, population-based studies are required.

## Trial registration

*PROSPERO review registration*: CRD42020162917 https://www.google.com/search?client=firefox-b-d&q=CRD42020162917.

## Introduction

Insecure parent-child attachment is associated with higher disease burden in adulthood [1,2]. As such, problems in parent-child interaction can act as a potential early observable risk factor, and target for early intervention [3]. Research has shown that parenting interventions have been found to be an effective method of improving parent-child interactions, emotional and behavioural adjustment of children and the psychosocial well-being of parents [4]. A previous study has shown that predictors of such difficulties in parent-child relationships can be identified within the first 10 months of life, within a general population sample, and there is a known association between infant experience, specifically these parent-child interactions, and later childhood psychopathology [5]. However, to date, this relationship has largely been investigated in high-risk populations, e.g., children diagnosed with autism or ADHD or parents with a psychiatric diagnosis. In several countries universal assessment of parent-infant interactions by primary healthcare professionals is mandatory but there is little to no guidance on the best methods to be used [6].

Several measures have been developed to assess parent-child relationships. These include parental self-report questionnaires and direct observation measures [7,8]. It has been argued that observational measures are preferable as they are less influenced by bias related to parental mood, expectations of intervention, or overestimation of change following intervention [9,10]. Observational measures are also particularly sensitive to changes in parent-infant interaction over time, much greater than those detected by parent self-reported measures [9]. A systematic review by Lotzin and colleagues identified over 500 observational tools used to assess parent-infant interaction, many of which were not published in peer-reviewed journal articles [7]. They report 24 observational tools used to measure parent-infant interaction, which were noted to be highly heterogeneous, some assessing direct behaviour and others evaluating broader concepts such as maternal sensitivity. Fourteen tools were designed to screen for potential problems requiring further evaluation and six aimed to identify problems in parent-infant interaction that needed intervention. Of relevance to this study, 71% were deemed suitable for low-risk community populations (63% for both high and low-risk samples and 8% specifically for low-risk) and 71% were suitable for use in either a home or clinical setting. Of interest, when considering the practicality of using such tools in clinical settings, time to complete the observations ranged from 1 to 45 minutes and time to score them ranged from 5 to 50 minutes. This paper highlighted the need for further research to identify which measures were associated with each child developmental outcome.

Gridley et al. examined the psychometric properties of outcome measures used in randomised controlled trials of parenting interventions [11]. They were only able to identify five

measures that met their criteria and noted the assessment of their validity to be poor (as assessed using COSMIN and modified Terwee checklists) [12,13]. These measures were the Attachment Q-Sort (AQS) [14], Communication and Symbolic Behavior Scales-Developmental Profile Behavior Sample (CSBS-DP) [15], Emotional Availability Scales (EAS) [16], Infant-Toddler Home Observational Measurement of the Environment ((IT-HOME) and Early Childhood Home Observational Measure of the Environment (EC-HOME) [17]. The AQS is a measure of attachment security, CSBS-DP measures early signs of communication delay, EAS measures adult sensitivity, structuring, non-intrusiveness and non-hostility, child responsiveness to adult and child involvement of the adult and the IT-HOME and EC-HOME observe parental responsivity, acceptance of the child, organization of the environment, learning materials, parental involvement, and variety within the home environment. Such measures could be helpful screening tools in primary care to aid the identification of children and families who would benefit from onward referral to infant mental health services. Primary care clinicians are particularly well placed to be involved in such assessment and screening as they have ongoing contact with almost all families with young children for a variety of reasons, including routine immunisations. Nevertheless, clinicians report a lack of training in parent-child observational assessment, despite a desire to acquire this competence [18]. Further research is needed before these measures could be considered for routine clinical use [19].

Regardless of the association between parent-child interaction and later childhood psychopathology in high-risk samples, and the number of observation tools currently available, to our knowledge, no review has yet synthesised the current evidence base in low-risk samples. Here, we define 'low-risk' samples as those recruited from population or community-based birth cohort data and not from populations known to have a diagnosis of psychiatric illness or developmental disorder, or at high genetic or environmental risk of these disorders. Despite a prior focus on targeting those deemed most 'at risk', it has been argued that effective research and health care provision must be universally developed and delivered with all in mind [20]. The term 'proportionate universalism' captures this need: "Focusing solely on the most disadvantaged will not reduce health inequalities sufficiently. To reduce the steepness of the social gradient in health, actions must be universal, but with a scale and intensity that is proportionate to the level of disadvantage" [21]. Therefore, this review will extend previous research and synthesise current findings on the observation tools available to predict later childhood psychopathology from parent-infant interactions in low-risk samples, where the evidence on whether to implement parent-child interaction assessments in routine care may be of use to all.

### Aims and objectives

The aim of this review is to bridge the gap within current literature by reporting the utility of various observational methods currently used to assess the parent-child relationship–throughout the first year of life–and report how such methods may be used to identify children—within low-risk samples—most at risk of developing later childhood psychopathology. This early screening method could then ensure appropriate onward referral to infant mental health services of children who present with a higher risk of developing a psychiatric disorder and encourage early intervention to better support families with young children.

### Methods

We followed PRISMA guidelines for systematic reviews (**S1 File**) [22]. The protocol for this review was registered with PROSPERO in advance of data extraction. The protocol ID is: CRD42020162917 and it can be found at: https://www.crd.york.ac.uk/prospero/.

A previous review shows that most studies examining the relationship between parent-infant interaction and later psychiatric diagnoses have focused on 'high-risk' cohorts or are disease specific [23]. 'High-risk' in this context would be studies exclusively examining cohorts with a sibling or parent with a mental illness or studies of low birth weight, premature infants, diagnosis of developmental disorder and those with other physical comorbidities. These are independent risk factors for developing mental illness [24]. As such, we have identified a gap within the current literature and wish to extend the research on the association between parent-infant interaction and later childhood psychopathology to include those from 'low-risk' populations. Positive and negative predictive values of observations vary markedly with prevalence and are thus very different in 'high-risk' and whole population samples. Therefore, data from 'high-risk' studies are likely to have limited value for primary care clinicians dealing mainly with unaffected individuals. Accordingly, we will examine population or birth cohort data that include a range of psychiatric diagnoses as outcome measures and exclude papers focusing on 'high-risk' groups. Psychiatric diagnoses were defined as any condition listed within the International Statistical Classification of Diseases or Diagnostic and Statistical Manual of Mental Disorders [25,26]. Birth cohort data also allow comparison of the utility of variables in prediction of different, potentially overlapping disorders [27]. We included case-control papers nested within population cohorts. The data from those studies presented samples enriched with cases to increase statistical power but was not taken specifically from high-risk groups so was considered valid for inclusion in this review.

To have most relevance to primary care clinicians, we limited the scope of this review to papers that assessed parent-child interaction through direct observation. This could include any measure of parent-infant interaction, such as level of maternal and infant speech, gaze or activity level and was not restricted to measures of parental sensitivity. As discussed previously, parent self-report measures are widely known to involve a high degree of bias and can be greatly impacted by the parent's mood at the time of assessment, so observations made by a neutral observer may have particular value [9,10]. This review has particular relevance to those working in primary care, where direct observation of parent-child interactions throughout routine care appointments is possible. Studies assessing parent-infant interaction up to the age of 13-months were included to encompass assessments over the first year of life, with a margin for error for those a little over 12 months. This is a period where families are often seen frequently for immunisations and other health checks, providing repeated opportunities for observation of interaction [28].

Eligibility criteria for inclusion were:

- Studies of the association between an independent observation of parent-infant interaction (children aged up to 13 months) and later childhood psychiatric diagnosis (occurring between ages 1–18 years inclusive)

- Birth cohorts or population–based samples (or case-control samples nested within a population study).

Exclusion criteria were:

- Non-peer-reviewed journal articles

- High-risk samples

- No discrete data on the assessment of infants under the age of 13 months

- Papers using parent report measure for the interaction assessment alone

- Outcome was not a psychiatric diagnosis or a symptom of mental illness

- Non-English language papers.

A systematic search was performed on EMBASE, CINAHL, PsycINFO, MIDIRS, Medline and Cochrane Library databases in May 2022. All available dates for each database were included during the search strategy, with no limits set on publication date. Reference lists of all included studies and reviews were also searched. Both Medical Subject Headings (MeSH terms) and keywords were used in the searches, and a full search strategy is shown in **S2 File**. One reviewer designed the search strategy and screened the results based on reading the titles and abstracts. The same reviewer then assessed the papers identified for full screening. Two other reviewers independently assessed half of the full text papers each for eligibility, blind to the other reviewer's decision. A third reviewer arbitrated on any disagreements. A data extraction form was created using the SIGN methodology checklist 3 (with irrelevant items removed) and Cochrane collaboration data extraction forms as a guide and is included as a **S3 File** [29,30]. Descriptive data on study characteristics—as well as details of the assessment, outcome measures and their results—and the statistical analysis performed were extracted. The data extraction form also incorporated a risk of bias assessment adapted from the SIGN checklist listed above. RevMan software from the Cochrane Collaboration was used to display the results of this assessment graphically [31].

A quantitative synthesis and meta-analysis were not possible given the highly heterogeneous methodologies and outcome measures used in the included papers. This also prohibited subgroup quantitative analysis. Therefore, a narrative synthesis was performed. Specific guidance developed by the Economic and Social Research Council was used to structure the synthesis and maintain methodological rigour [32].

## Results

A systematic search was performed on six databases. 5223 papers were identified from EMBASE, 3768 were identified from CINAHL, 3335 from Cochrane Library, 1145 from PsycINFO, 1693 from MIDIRS and 8068 from Medline, producing 23,232 hits in total. An additional six articles were identified through reviewing reference lists of full text articles. An initial screen identified 3187 duplicates, which were removed, leaving 20,051 articles. The titles and abstracts were then reviewed to assess suitability, and after exclusion of articles unrelated to our research question, 30 articles remained. 4 further articles were then excluded as they were not from a peer reviewed journal, and one excluded as it was not in English. This left 25 articles for which the full text was reviewed. On reviewing the full texts, a further 16 were excluded, leaving 9 articles for inclusion in our study. This is illustrated in the PRISMA flowchart below (Fig 1).

### Study characteristics

Study characteristics and a summary of the key aspects of methodology of the nine included papers are included below in Table 1 [33–41]. Of note, five papers from the same research team were all based on the English ALSPAC (Avon Longitudinal Study of Parents and Children) Child in Focus sub-cohort. These papers all utilised the same case-control sample (with slightly differing numbers due to varying exclusion criteria), nested within the ALSPAC community cohort. We included all five papers within our review as they examined different assessment variables, using differing measures. As such, they contained sufficiently different data to merit inclusion. Details of study size, gender, ethnicity, and socio-demographic information are displayed where available but was not complete for all studies. The ALSPAC cohort is noted to be slightly more affluent than the general population it was taken from, with slightly fewer families from ethnic minority backgrounds [42]. Specific sociodemographic data for the sub-cohort used in papers 1–2 and 5–7 are not available. Likewise the CCC2000 (Copenhagen

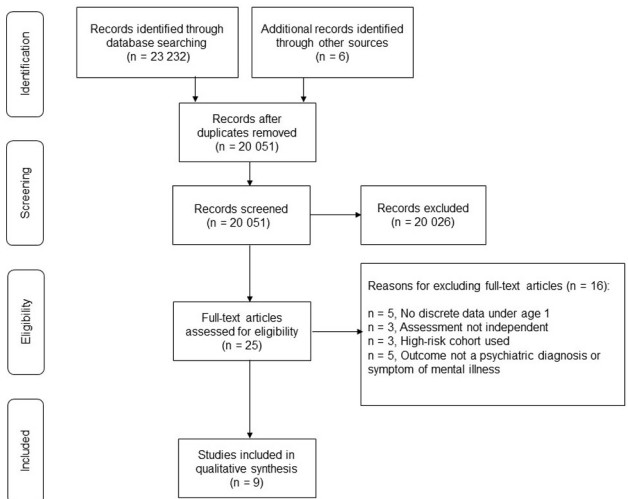

**Fig 1. PRISMA flowchart illustrating papers identified and included in the study.**

Child Cohort 2000) cohort (paper 3) was slightly more affluent than the general population, but also had a higher number of immigrant families [43]. Two studies (papers 8 and 9) were based on cohorts from the USA and had a higher proportion of families from African American and Hispanic groups.

## Assessment of risk of bias

A risk of bias assessment was completed for each study and the results illustrated in the figure below (Fig 2). A minus sign indicates the potential for bias in an element of study methodology, whilst a plus sign is used where adequate steps appear to have been taken to minimise bias. Blank cells are shown where there was insufficient detail to make a judgement. No papers were excluded based on this assessment. Some papers were of lower quality but were included due to the paucity of data available. This was considered when drawing conclusions from the included studies.

## Study methodology

The studies included in this review were very heterogeneous in their methodology. A quantitative analysis was not possible; therefore, a qualitative analysis was performed. The key aspects of methodology are listed in Table 1, including data on participants, exposures, and outcomes. Although six papers used the same outcome variable and outcome measure, it was not felt to be appropriate to combine these data as five of the six papers utilised the same cohort sample. The remaining papers all differed substantially in their assessment variable, assessment/outcome age or outcome measure and as such, were not considered suitable to combine for statistical analysis.

## Independent observation of parent-child interaction

Five papers (1, 2, 5, 6 and 7) used ALSPAC cohort participants. A subgroup of this cohort attended additional assessments, entitled 'Children in Focus' clinics. This included the Thorpe Interactive Measure (TIM) at one session [44]. This was a videotaped interaction between a parent and a child at age 12 months, seated on a sofa looking at a picture book. The above studies included in this paper were based on a nested case-control sample taken from this sub-

**Table 1. Study characteristics and methodology of included papers.**

| Study | Title | Author | Sample details | Population demographics | Assessment age | Exposures | Measure | Outcome age | Outcomes |
|---|---|---|---|---|---|---|---|---|---|
| 1 | Association between parent-infant interactions in infancy and disruptive behaviour disorders at age seven: a nested, case-control ALSPAC study | (Puckering et al., 2014) [41] | Nested case-control group within community sample: 54 cases, 106 controls (49 F, 111 M) | 97.8% White Caucasian, 79.4% married couple, 79.1% homeowners | 12 months | Maternal-Infant interaction | Mellow Parenting Observational System | 90 months | Psychiatric diagnoses assessed by DAWBA*a |
| 2 | Can psychopathology at age 7 be predicted from clinical observation at one year? Evidence from the ALSPAC cohort | (Allely et al., 2012) [34] | Nested case-control group within community sample: 60 cases, 120 controls | 97.8% White Caucasian, 79.4% married couple, 79.1% homeowners | 12 months | Likelihood of becoming a 'case' | Clinical judgement | 90 months | Psychiatric diagnoses assessed by DAWBA |
| 3 | Infancy predictors of hyperkinetic and pervasive developmental disorders at ages 5–7 years: results from the Copenhagen Child Cohort CCC2000 | (Elberling et al., 2014) [37] | Community sample: N 1585 (M:F 1:1) | 93% children born at term, 79.2% had both parents born in Denmark, 94% parents living together | 1–5 weeks, 2–3 months, 4–6 months, 8–10 months | Maternal-Infant Interaction | Manualised | 5–8 years | Psychiatric diagnoses assessed by DAWBA |
| 4 | Infant and dyadic assessment in early community-based screening for autism spectrum disorder with the PREAUT grid | (Olliac et al., 2017) [40] | Community sample, N 12,179 (49% F, 51% M) | None collected | 4 and 9 months | Early social interaction | PREAUT*b grid | 3 years | CARS*c, clinical assessment of autism (ADIR*d/ADOS*e in some) |
| 5 | Parent-infant vocalisations at 12 months predict psychopathology at 7 years | (Allely et al., 2013) [33] | Nested case-control group within community sample: 58 cases, 111 controls (53 F, 116 M) | 97.8% White Caucasian, 79.4% married couple, 79.1% homeowners | 12 months | Vocalisations of Parents and Infants | Vocalisation rate and frequency using PRAAT software | 90 months | Psychiatric diagnoses assessed by DAWBA |
| 6 | Prediction of 7-year psychopathology from mother-infant joint attention behaviours: a nested case-control study | (Allely et al., 2013) [35] | Nested case-control group within community sample: 53 cases, 106 controls (48 F, 111 M) | 97.8% White Caucasian, 79.4% married couple, 79.1% homeowners | 12 months | Mother-Child interaction | Joint Attention | 90 months | Psychiatric diagnoses assessed by DAWBA |
| 7 | Predictors of diagnosis of child psychiatric disorder in adult-infant social-communicative interaction at 12 months | (Marwick et al., 2013) [39] | Nested case-control group within community sample: 60 cases, 120 controls (56 F, 122 M) | 97.8% White Caucasian, 79.4% married couple, 79.1% homeowners | 12 months | Mother-Child interaction | Holistic measure using 8 categories | 90 months | Psychiatric diagnoses assessed by DAWBA |

*(Continued)*

**Table 1.** (Continued)

| Study | Title | Author | Sample details | Population demographics | Assessment age | Exposures | Measure | Outcome age | Outcomes |
|---|---|---|---|---|---|---|---|---|---|
| 8 | Temperament and Parenting during the first year of life predict future child conduct problems | (Lahey et al., 2008) [38] | Community sample: N 1519 (47.4% F, 52,6% M) | 18.9% Hispanic, 27.1% African American, 54.0% European American | 0–11 months | Maternal responsiveness | HOME-SF*f | 4–13 years | Conduct problems assessed by the Behaviour problem index |
| 9 | Predicting children's separation anxiety at age 6: The contributions of infant-mother attachment security, maternal sensitivity and maternal separation anxiety | (Dallaire & Weinraub, 2005) [36] | Community sample: N 99 (48% F, 52% M) | 67 European American, 26 African American, 2 Hispanic American, 4 Asian American | 6 months | Maternal sensitivity | Bespoke measure | 6 years | Child separation anxiety assessed by the Modified Child Puppet Interview |

*a Development and Wellbeing Assessment.

*b Programme de Recherches et d'Etudes sur l'AUTisme.

*c Childhood Autism Rating Scale.

*d Autism Diagnostic Interview-Revised.

*e Autism Diagnostic Observation Schedule.

*f Home Observation Measurement of the Environment-Short Form.

cohort. Sixty of these 1240 infants in the sub-cohort were later assessed as having a probable psychiatric diagnosis by a child psychiatrist based on a Development and Wellbeing Assessment (DAWBA) completed by parents and teachers of the children at age 91 months [45]. One hundred and twenty sex-matched controls were then randomly selected from the non-case videos.

Paper 1 included 160 of these videos, where the mother was identified as the lead caregiver. It excluded videos where the father was the lead caregiver, to avoid a potential confounding factor. The Mellow Parenting Observational System [46] was then used to assess maternal-infant interactions on the TIM videotapes, with raters blind to case or control status. This recorded the rate of positive and negative interactions across six dimensions. Inter-rater reliability was also assessed and found to be moderately reliable with an inter-class correlation of 53%.

Paper 2 included all 60 cases and 120 controls in their study. Two groups of experienced child mental health clinicians, blind to case control status were asked to make a clinical judgement on whether each child was likely to be a case or control, and what their diagnosis might be. Four videos were then excluded due to poor quality. Different rating scales were used by the two groups. Inter-rater agreement was calculated using kappa statistics and found to be poor (less than 0.4).

Paper 5 excluded seven videos where the father was the sole caregiver present and four due to poor quality, including 169 videos in total. Vocalisations by both the mother and the infant were analysed using PRAAT software (http://www.fon.hum.uva.nl/praat/) to determine frequency and duration of vocalisation for each individual. This was done blind to their diagnostic status and a subset was analysed by two raters to assess reliability. Maternal vocalisation rates and frequencies were measured reliably (interclass correlation coefficients of 83% and 88%), as were infant vocalisation frequencies (85%), but infant vocalisation rates less so (61%).

Paper 6 utilised 159 videos, excluding nineteen where the father was the lead caregiver, and two due to poor quality. These were then examined using Noldus Observer software (https://

| | Participant selection (selection bias) | Incomplete outcome data (attrition bias) | Unvalidated assessment tool | Blinding of outcome assessment (detection bias) | Confounders | Unvalidated outcome tool |
|---|---|---|---|---|---|---|
| Allely 2012 | + | + | - | + | + | + |
| Allely 2013 (a) | + | + | + | + | + | + |
| Allely 2013 (b) | + | + | + | + | + | + |
| Elberling 2014 | + | + | - | | + | + |
| Horvath Dallaire 2005 | + | + | | | | |
| Lahey 2008 | + | | + | + | + | |
| Marwick 2013 | + | + | + | + | + | + |
| Olliac 2017 | + | - | - | | - | |
| Puckering 2014 | + | + | + | + | + | + |

**Fig 2. RevMan risk of bias illustration for included studies.**

www.noldus.com/observer-xt) by independent observer's blind to all other details. Joint attention was examined based on shared looks per minute between mother and infant, percentage of time spent in shared looks, and number of periods of shared attention per minute. A subset was coded by two raters to assess reliability. Kendall's $\tau$ were consistent with high inter-rater reliability (0.83, 1.00 and 0.97) for the three measures used.

Paper 7 excluded two videos due to poor quality, so included 178 parent-infant interactions in their study. Assessors blind to the child's later case control status scored each dyad on a scale of 1–5 indicating extent of occurrence within eight categories: well-being, contingent responsiveness, cooperativeness, involvement, activity, playfulness, fussiness, and speech. This system was adapted from categories of interpersonal interaction developed by Marwick and Trevarthen previously [47,48]. Inter-rater reliability tests using weighted kappa statistics and rank correlations were carried out on a sample of 29 videos and found to be very varied (-0.04–0.73). The authors report low weighted kappa statistics for scales with little variation.

Paper 3 utilised data from a subset of children from the CCC2000. This cohort participated in two phases of assessment. At phase 1 the Strengths and Difficulties questionnaire (SDQ) [49] was completed and at phase 2 the DAWBA. For phase 2, all children with SDQ scores indicating a possible mental disorder were invited to complete a DAWBA, as were a random sample from the entire birth cohort. 77% participated and form the sample for Elberling's study. Assessments of these children were performed by community health nurses in the families' homes, using a standardised record. At each visit nurses made an assessment of the

mother-child relationship (amongst other measures) based on direct observation [5]. This included the mother's expectations and handling of the child, the affective involvement and synchronicity of enjoyment between the pair. This was categorised as 'not normal' if the nurse did not consider the interaction was within the normal range at one or more visits.

Paper 4 was a large multicentre study across 10 PMI centres (Centre de Protection Maternelle et Infantile) in France. 12,179 infants who were registered at these centres between September 2005 and November 2011 agreed to take part. Infants were screened using the PREAUT (Programme de Recherches et d'Etudes sur l'AUTisme) grid at age 4 and 9 months [50]. The grid is scored by a trained paediatrician during a visit with the family and evaluates the infant's ability to engage spontaneously in synchronous and joyful exchanges. Pathological thresholds were adjusted for this study based on the results of a preliminary exploratory study in the general population. Inter-rater reliability was not calculated within this study. 60% of infants were lost to follow up before further screening at age two and 55% of those who screened positive at any stage were lost to follow up before diagnostic assessment could take place. All children who screened positive were offered diagnostic follow up between ages 3 and 4 using a range of different measures including the Autism Diagnostic Observation Schedule (ADOS) [51] or Autism Diagnostic Interview-Revised (ADIR) [52] (performed by specialist services) and Childhood Autism Rating Scale (CARS) [53]. One of the families receiving a diagnosis of Autism Spectrum Disorder refused specialist assessment however, so their diagnosis was based on the findings of the research team. All diagnoses were based on ICD-10 criteria [25].

Paper 8 utilised the Children of the National Longitudinal Survey of Youth (CNLSY) sample. This was based on a United States nationally representative household sample who then went on to have children. This sample was enriched to oversample African American and Hispanic mothers. Their final analysis was based on a subset of this sample, for which there was an interviewer rating of parent-child interaction (and infant temperament) in the first year of life, and at least two follow up assessments of child conduct problems later in childhood. One hundred and seventy-seven parents were excluded as they did not provide details on family income, which was considered an important potential confounding factor. Infant sex, family income, mother's age and ethnicity were controlled for.

Parenting was assessed using the Infant/Toddler Home Observation for Measurement of the Environment–Short Form (HOME-SF) [16]. This includes an interviewer rating of maternal responsiveness. Conduct problems were then assessed later in childhood using the Behavioural Problem Index (BPI), completed by the mother [54]. Seven items from the BPI that were related to symptoms of conduct disorder were then selected for data analysis.

Finally, paper 9 involved a sample of families from the Temple University site of the NICHD Study of Early Childcare. Of the 136 eligible families, 99 agreed to be involved in their study. No significant differences between participating and non-participating families were found. This paper reports on their findings related to maternal sensitivity during a 15-minute free play interaction at the family home at age six months. Trained coders rated videotapes of the interactions and 20% of videos were assigned to two coders to assess inter-rater reliability. Intra-class correlation coefficient was found to be 0.87.

Sensitivity to non-distress (defined as: the extent to which the mother responds promptly and appropriately to the child's social gestures, expressions and signals, and the extent to which the mother is child-centred), maternal intrusiveness and maternal positive regard for the child were rated on a four-point scale developed by the authors for this study and then a composite score generated. Child separation anxiety was then assessed by a trained research assistant using the Child Puppet Interview [55], modified to include a scale of eight items related to separation anxiety.

## Narrative synthesis of results

To answer our question of whether childhood psychopathology–within low-risk samples–could be predicted from observing parent-child interaction, the results of the studies were grouped according to categories of psychiatric diagnoses. This was considered to allow for better comparison of the effectiveness of various measures rather than grouping all the results from each paper together. Statistically significant results (p<0.05) are summarised in Table 2 below–no correction is applied to compensate for multiple outcomes. Full results are displayed in a S4 File.

Paper 2 reported that specialist clinicians were unable to predict overall case status, except possibly in the instance of inattentive ADHD. Likewise, joint attention was not significantly associated with any diagnosis other than childhood oppositional defiant or conduct disorder. Child vocalisation frequency was only significantly associated with disruptive behaviour disorders. Adult and infant cooperativeness was not significantly associated with any later diagnosis of childhood mental illness. The manualised assessment of mother-infant relationship in paper 3 was only significantly associated with hyperkinetic disorders once adjustments had been made for confounding factors. Adjusting for maternal psychological problems and infant development reduced the strength of associations found across all diagnostic domains. Therefore, these did not appear to be particularly sensitive measures for predicting later childhood psychopathology.

Paper 1 looked at the association of their measure with overall psychopathology, emotional disorders, disruptive behaviour disorders and oppositional defiant/conduct disorders and found a relatively strong inverse association of positive parenting behaviours with each of these diagnostic groups. Parental vocalisation frequency was associated with each diagnostic group except ADHD or ASD. Similarly, adult speech and activity was significantly associated with all diagnostic groups except a later diagnosis of ASD. Taken together, level of maternal activity (vocalisation, physical activity, positive parenting, and shared attention) appears associated with later childhood psychopathology.

No included studies found their assessment of parent-infant interaction to be significantly associated with a later diagnosis of ASD, and paper 4—using the PREAUT grid—found their measure could contribute to the early diagnosis of ASD but on its own did not have a strong positive predictive value.

## Discussion

Our review only included studies that used population based or birth cohort data. Studies based on high-risk samples were not included. In many countries, non-specialists in child development—such as Health Visitors and General Practitioners—would be the only clinicians regularly seeing infants in the first year of their life. Here, they have the routine opportunity to observe parent-child interactions and we consider our findings to be of most relevance to these clinicians. Although reducing the number of studies available for analysis, the exclusion of studies relying solely on parent self-report aligns with the current focus in primary care research. Nevertheless, the small number of eligible studies is remarkable [23].

Our review suggests that assessments of parent-infant interaction, and in particular levels of maternal activity, may be an early observable risk factor for later childhood psychopathology in low-risk, community-based samples. There may be both genetic and parenting-based explanations for these associations. Despite there being very little data available on the validity of the measures used by the included studies, other existing validated measures are noted to require intense training to be used effectively [56]. Training for primary care clinicians in assessing parent-child interactions is not currently standardised and as such can prove difficult

**Table 2. Summary of statistically significant associations between assessments of parenting and childhood psychopathology from papers included in the systematic review.**

| Study | Outcome | Variable | Statistical Test | Result | Magnitude of association between observation and outcome |
|---|---|---|---|---|---|
| 1 | Psychopathology | Positive parent-infant interaction | Odds ratio (95% confidence intervals) | 0.85 (0.74–0.96) | An increase of one positive interaction per minute (from a sample mean 6.2 [SD 3.3] per minute) predicted a 15% reduction in the odds of diagnosis |
| 5 | Psychopathology | Parent vocalisation frequency | Odds ratio (95% confidence intervals) | 0.69 (0.52–0.90) | Control median of 18 (IQR 13.5–23) compared with case median of 14.5 (IQR 11.5–18.5) vocalisations per minute, approx. effect size 0.4. |
| 7 | Psychopathology | Adult speech and activity | Odds ratio (95% confidence intervals) | 0.57 (0.40–0.80) | For every 1 standard deviation increase in the combined adult verbal/activity score the odds of having a diagnosis reduced by 43%. |
| 1 | Emotional disorders | Positive parent-infant interaction | Odds ratio (95% confidence intervals) | 0.82 (0.66–0.98) | An increase of one positive interaction per minute (from a sample mean 6.2 [SD 3.3] per minute) predicted an 18% reduction in the odds of diagnosis |
| 5 | Emotional disorders | Parent vocalisation frequency | Odds ratio (95% confidence intervals) | 0.63 (0.42–0.92) | Control median of 18 (IQR 13.5–23) compared with case median of 14 (IQR 10–19.5) vocalisations per minute, approx. effect size 0.4. |
| 7 | Emotional disorders | Adult speech and activity | Odds ratio (95% confidence intervals) | 0.51 (0.31–0.80) | For every 1 standard deviation increase in the combined adult verbal/activity score the odds of having a diagnosis reduced by 49%. |
| 7 | Anxiety disorders | Adult speech and activity | Odds ratio (95% confidence intervals) | 0.47 (0.28–0.75) | For every 1 standard deviation increase in the combined adult verbal/activity score the odds of having a diagnosis reduced by 53%. |
| 9 | Child Separation anxiety | Maternal sensitivity | Pearson product moment correlation coefficient | -0.29 (p<0.01) | Children's reports of separation anxiety at age 6 years were negatively correlated with ratings of mothers' sensitivity at 6. However regression model showed no significant contribution of maternal sensitivity at 6 months to separation anxiety at 6 years. |
| 1 | Disruptive behaviour disorders | Positive parent-infant interaction | Odds ratio (95% confidence intervals) | 0.84 (0.71–0.97) | An increase of one positive interaction per minute (from a sample mean 6.2 [SD 3.3] per minute) predicted a 16% reduction in the odds of diagnosis |
| 5 | Disruptive behaviour disorders | Child vocalisation frequency | Odds ratio (95% confidence intervals) | 1.77 (1.07–3.05) | Control median of 4.7 (IQR 2.5–7.5) compared with case median of 5.7 (IQR 3.7–8.7) vocalisations per minute, approx. effect size 0.2. |
| 5 | Disruptive behaviour disorders | Parent vocalisation frequency | Odds ratio (95% confidence intervals) | 0.68 (0.47–0.94) | Control median of 18 (IQR 13.5–23) compared with case median of 14 (IQR 10.5–18) vocalisations per minute, approx. effect size 0.4. |
| 7 | Disruptive Behaviour disorders | Adult speech and activity | Odds ratio (95% confidence intervals) | 0.53 (0.34–0.79) | For every 1 standard deviation increase in the combined adult verbal/activity score the odds of having a diagnosis reduced by 47%. |
| 1 | Oppositional defiant & Conduct disorders | Positive parent-infant interaction | Odds ratio (95% confidence intervals) | 0.81 (0.65–0.97) | An increase of one positive interaction per minute (from a sample mean 6.2 [SD 3.3] per minute) predicted a 19% reduction in the odds of diagnosis |
| 5 | Oppositional defiant & Conduct disorders | Parent vocalisation frequency | Odds ratio (95% confidence intervals) | 0.64 (0.41–0.94) | Control median of 18 (IQR 13.5–23) compared with case median of 13.5 (IQR 9.5–18) vocalisations per minute, approx. effect size 0.5. |
| 6 | Oppositional defiant & Conduct disorders | Parent-infant shared look rate | Odds ratio (95% confidence intervals) | 1.5 (1.0–2.3) | For every 1 standard deviation increase in the shared look rate the odds of having a diagnosis increased by 50%. |
| 7 | Oppositional defiant & Conduct disorders | Adult speech and activity | Odds ratio (95% confidence intervals) | 0.50 (0.31–0.80) | For every 1 standard deviation increase in the combined adult verbal/activity score the odds of having a diagnosis reduced by 50%. |
| 7 | Conduct disorder | Adult speech and activity | Odds ratio (95% confidence intervals) | 0.37 (0.12–0.98) | For every 1 standard deviation increase in the combined adult verbal/activity score the odds of having a diagnosis reduced by 63%. |
| 2 | Inattentive Attention Deficit Hyperactivity Disorder | Likelihood of being a 'case' | Fisher's exact test | P = 0.039 | Overall sensitivity 0.80 and specificity 1.00 for prediction of inattentive ADHD by at least one rater. |

*(Continued)*

**Table 2.**  (Continued)

| Study | Outcome | Variable | Statistical Test | Result | Magnitude of association between observation and outcome |
|---|---|---|---|---|---|
| 3 | Attention Deficit Hyperactivity Disorder & Hyperkinetic disorder | Maternal-infant interaction | Odds ratio (95% confidence intervals) | 5.20 (1.55–17.47) | Overall sensitivity 0.32 and specificity 0.93 for prediction of ADHD from any clinical observation of abnormal maternal-infant interaction at any one of four observations before 10 months |
| 7 | Attention Deficit Hyperactivity Disorder & Hyperkinetic disorder | Adult speech and activity | Odds ratio (95% confidence intervals) | 0.42 (0.23–0.70) | For every 1 standard deviation increase in the combined adult verbal/activity score the odds of having a diagnosis reduced by 58%. |
| 7 | Inattentive Attention Deficit Hyperactivity Disorder | Adult speech and activity | Odds ratio (95% confidence intervals) | 0.19 (0.04–0.56) | For every 1 standard deviation increase in the combined adult verbal/activity score the odds of having a diagnosis reduced by 81%. |

to perform reliably [57]. Only papers 3 and 4 utilised community clinicians to assess participants. Therefore, further research is needed in clinical—as opposed to research—settings to determine which measures can be pragmatically used as general screening measures.

Thompson et al. [56] describe a simplified tool, Child and Adult Relationship Observation (CARO) based on the Mellow Parenting Observational System utilised in Paper 1. As such, it is possible that evidence from more complicated assessment tools, such as the assessment of vocalisations described in paper 5 and of joint attention in paper 6—which currently require specialist software, may guide further enquiry and development of future streamlined, novel tools. Paper 2 utilised clinical judgement, with no training required, and so the results with relation to prediction of inattentive ADHD show clinical utility. Papers 7 and 9 describe holistic measures assessing interaction across several categories, but do not appear to be validated. Data were not available on the training required to complete these assessments, but no specialist equipment was used. Paper 8 utilised the Home Observation Measurement of the Environment-Short Form, which is noted to take half as long to complete as the original HOME Inventory [16].

Interrater reliability measures are discussed in papers 1, 2, 6 and 9. Measures of the rate of positive interaction in paper 1 were found to be moderately reliable with an inter-class correlation of 53%. Measurements of the rate of negative interactions were also reliable, with a correlation of 0.6 using Kendall's τ. However, agreement between raters using clinical judgement in paper 2 was generally poor for all assessments, with all Kappa statistics being less than 0.4. In paper 6, agreement between raters was high for shared look rate, with estimates of Kendall's τ 0.83. Finally, paper 9 also found good inter-rater reliability, with an intra-class correlation coefficient of 0.87, for their assessments at 6 months of age.

When examining the ability of the reviewed measures to detect different childhood disorders, our findings were similar to those reported by Baker et al.: maternal sensitivity does not appear to be strongly associated with later development of autism spectrum conditions [58]. Therefore, it is possible that disordered parent-child interaction is not a particularly sensitive indicator of later autism spectrum disorders. However, studies have suggested that if those at risk can be identified, parenting interventions can reduce both internalizing and externalizing disorders in children, highlighting the importance of ongoing research in this area [59,60].

## Strengths and limitations

The main strength of our review is the high clinical relevance. The use of community-based samples makes the data more generalizable, though all but two papers were based in Europe. As most current literature within this field has a focus on observing parent-child interactions in high-risk samples (e.g., children diagnosed with ADHD/autism or a parent with a psychiatric diagnosis), our findings extend the evidence base for an association between parent-child

interactions and later psychopathology in the wider population. However, this review was only able to identify nine research papers that were directly relevant to our research question, and five of these used the same participants. This meant we could not give the same weight to our findings as we could have done if the samples were different.

Similar to the findings by Lotzin et al. [7], the results were highly heterogeneous, meaning it was not possible to perform any quantitative analysis. It is possible this high heterogeneity exists as assessing parent-infant interactions is not currently a routine practice in most countries, limiting the potential data available for analysis. This limitation means our conclusions must be interpreted with caution, as they reflect the authors' interpretation of the results. The advantage of proceeding with a qualitative synthesis of our results, however, is that it pulls together the existing divergent evidence and highlights a clear case for further research in this area.

The heterogeneity of study designs and types of outcome measure, the lack of prior validation of most of the observational tools and varying approaches to sampling make it difficult to articulate the magnitude of the associations between observed behaviours and psychiatric outcomes in a way that could be directly applied to clinical practice in the community. In no case was it possible to state a simple 'effect size' for these associations but it was possible to derive estimates of sensitivity and specificity for some measures. The data in Table 2 indicate that some observations are likely to be particularly useful to primary care clinicians. For example, the use of professional judgement on the quality of parent-infant interaction for predicting ADHD and the rate of child vocalisation frequency in relation to disruptive behaviour problems.

Only published results from peer reviewed journals were included, so it is possible there is a degree of reporting bias in our review. Sources of potential bias in the literature were also identified. It should be noted that the two American studies had markedly different representations from Black and Minority Ethnic groups than the other studies, which had largely White Caucasian populations. Paper 8 controlled for ethnicity but paper 9 did not describe the variables they controlled for in their analysis. Therefore, this is a source of potential bias as parenting is known to vary with different cultural backgrounds [61]. The papers utilising a sample from the ALSPAC cohort were all based on a nested case-control design. As such, the results from these enriched samples may not accurately reflect the odds ratios in the general population.

As noted, the ALSPAC and CCC2000 cohorts were all slightly more affluent than the general populations they were taken from, potentially making these results less applicable to populations in more socio-economically deprived areas. Social class or family income was not included as a confounding factor in papers 1 (however, maternal education attainment was), 2, 4 or 7. Maternal age at birth of the child and infant weight at birth were also not considered in the analysis in papers 2, 4 or 7. These were both found to be significant factors in studies 1 and 5. Only papers 1, 5 and 6 considered maternal mental health as a potential confounding factor.

The ALSPAC studies used a sub sample of participants from the Children in Focus clinics, and only included families for which there were full data available. All other papers had a degree of drop out and this tends to result in under-representation of families with children with behavioural problems [62]. This was significant in paper 4 where the authors purposefully obtained a large sample to account for this. However, they did not make any analysis of the potential differences in baseline characteristics of those who participated compared to those who did not, perhaps leading to bias in their results. The remaining papers adjusted for differences between those who remained and those who dropped out in their analyses.

## Conclusions

This review synthesised the current observational methods used to assess parent-child interaction and its association with later childhood psychopathology. Unlike previous research, this

review included studies assessing parent-infant interactions in low-risk, community-based samples, helping to bridge the gap and extend the generalisability of our findings. Independent assessment of parent-child interactions in the first year of life appears to be a promising early indicator of later childhood psychopathology. A range of tools are being developed, which could potentially be incorporated into routine Health Visitor, Primary Care Paediatrician or General Practitioner reviews of infants. This could allow early identification of families at risk, so they can be provided with support in their parenting and more intensive follow up. In turn, this may potentially reduce later childhood psychopathology, but clinical trials are needed in this area. Some measures appeared more sensitive than others, and a comparative study looking at two or more measures would be of value in determining which might have greatest clinical utility.

## Supporting information

**S1 File. PRISMA 2009 checklist MS-Word.**
(DOC)

**S2 File. Electronic search strategy.**
(DOCX)

**S3 File. Data extraction form.**
(RTF)

**S4 File. Full results extracted from included papers.**
(DOCX)

## Acknowledgments

### Declarations

We wish to thank Lyn Mair, Clinical Liaison Librarian for NHS Grampian for her assistance in developing the literature search strategy.

## Author Contributions

**Conceptualization:** Elena McAndie, Philip Wilson, Lucy Thompson.

**Data curation:** Elena McAndie, Charlotte Alice Murray, Lucy Thompson.

**Formal analysis:** Elena McAndie, Lucy Thompson.

**Methodology:** Lucy Thompson.

**Project administration:** Elena McAndie, Charlotte Alice Murray, Philip Wilson, Lucy Thompson.

**Supervision:** Philip Wilson, Lucy Thompson.

**Visualization:** Elena McAndie.

**Writing – original draft:** Elena McAndie, Philip Wilson, Lucy Thompson.

**Writing – review & editing:** Charlotte Alice Murray, Philip Wilson.

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
