## [Decision Letter · Decision Letter 0]

30 Aug 2022

PONE-D-22-23465Parent-infant observation for prediction of later childhood psychopathology in ‘low-risk’ populations: A Systematic ReviewPLOS ONE

Dear Dr. Thompson,

Thank you for submitting your manuscript to PLOS ONE. After careful consideration, we feel that it has merit but does not fully meet PLOS ONE’s publication criteria as it currently stands. Therefore, we invite you to submit a revised version of the manuscript that addresses the points raised during the review process.

One expert in systematic review has provided professional comments for your to improve the submission. Please take the advice seriously in your revision. Additionally, please take the following considerations.1. The title, Abstract, and Introduction do not clearly explain what 'low-risk' populations are. I think that the authors should make this clear throughout.2. I would expect to see the authors provide clear information regarding PECO in the Methods section.3. I am not sure why the authors want to mention ACEs in the Introduction. Apparently, ACEs are not the focus of the present systematic review, nor did the authors discuss the ACEs among parent-child interaction. 

We look forward to receiving your revised manuscript.

Kind regards,

Chung-Ying Lin

Academic Editor

PLOS ONE

Journal Requirements:

"The study was funded by departmental resources."

"I have read the journal's policy and the authors of this manuscript have the following competing interests: PW is a co-author on five of the papers included in the analysis of this systematic review."

4. PLOS requires an ORCID iD for the corresponding author in Editorial Manager on papers submitted after December 6th, 2016. Please ensure that you have an ORCID iD and that it is validated in Editorial Manager. To do this, go to ‘Update my Information’ (in the upper left-hand corner of the main menu), and click on the Fetch/Validate link next to the ORCID field. This will take you to the ORCID site and allow you to create a new iD or authenticate a pre-existing iD in Editorial Manager. Please see the following video for instructions on linking an ORCID iD to your Editorial Manager account: https://www.youtube.com/watch?v=_xcclfuvtxQ.

Reviewers' comments:

Reviewer's Responses to Questions

**Comments to the Author**

1. Is the manuscript technically sound, and do the data support the conclusions?

Reviewer #1: Yes

2. Has the statistical analysis been performed appropriately and rigorously? 

Reviewer #1: Yes

3. Have the authors made all data underlying the findings in their manuscript fully available?

Reviewer #1: Yes

4. Is the manuscript presented in an intelligible fashion and written in standard English?

Reviewer #1: Yes

5. Review Comments to the Author

Reviewer #1: This study aimed to assess the utility of current methods used to observe parent-child interactions – within the first year of life – and their ability to screen and identify children from low-risk samples most at risk of developing childhood psychopathology.

I agree with most of the methodology that the authors used for this study. I have only a few points to consider:

1. The authors did not mention who screened the title/abstract and was this done by different reviewers independently?

2. The authors can elaborate more on the risk of bias assessment tools used for this study. Where does it come from and did they make adaptations?

3. It is unfortunately that these authors were not able to conduct quantitative synthesis owing to the limited numbers of studies included. The authors may discuss the appropriateness of inclusion/exclusion defined for this study. Why excluding parent-reported measures? Why excluding high risk samples? I understand that this cannot be changed at the current stage. Would a review with broader inclusion criteria needed in the future to examine this research question? And then the authors can carefully examine each subgroup within such a review.

6. PLOS authors have the option to publish the peer review history of their article (what does this mean?). If published, this will include your full peer review and any attached files.

Reviewer #1: No

---

## [Author Response · Author response to Decision Letter 0]

29 Nov 2022

Please see the cover letter and response to reviewers.

---

## [Decision Letter · Decision Letter 1]

12 Dec 2022

Parent-infant observation for prediction of later childhood psychopathology in community-based samples: A Systematic Review

PONE-D-22-23465R1

Dear Dr. McAndie,

We’re pleased to inform you that your manuscript has been judged scientifically suitable for publication and will be formally accepted for publication once it meets all outstanding technical requirements.

Kind regards,

Chung-Ying Lin

Academic Editor

PLOS ONE

Additional Editor Comments (optional):

Reviewers' comments:

Reviewer's Responses to Questions

**Comments to the Author**

1. If the authors have adequately addressed your comments raised in a previous round of review and you feel that this manuscript is now acceptable for publication, you may indicate that here to bypass the “Comments to the Author” section, enter your conflict of interest statement in the “Confidential to Editor” section, and submit your "Accept" recommendation.

Reviewer #1: All comments have been addressed

2. Is the manuscript technically sound, and do the data support the conclusions?

Reviewer #1: Yes

3. Has the statistical analysis been performed appropriately and rigorously? 

Reviewer #1: Yes

4. Have the authors made all data underlying the findings in their manuscript fully available?

Reviewer #1: Yes

5. Is the manuscript presented in an intelligible fashion and written in standard English?

Reviewer #1: Yes

6. Review Comments to the Author

Reviewer #1: (No Response)

7. PLOS authors have the option to publish the peer review history of their article (what does this mean?). If published, this will include your full peer review and any attached files.

Reviewer #1: No

---

## [Editor Report · Acceptance letter]

19 Dec 2022

PONE-D-22-23465R1 

Parent-infant observation for prediction of later childhood psychopathology in community-based samples: A Systematic Review 

Dear Dr. McAndie:

I'm pleased to inform you that your manuscript has been deemed suitable for publication in PLOS ONE. Congratulations! Your manuscript is now with our production department. 

Kind regards, 

on behalf of

Dr. Chung-Ying Lin 

Academic Editor

PLOS ONE